# Replicating Softmax Deep Double Deterministic Policy Gradients

## Reproducibility Summary

**Scope of Reproducibility**

We attempt to reproduce Pan et al. [25] claim that Softmax Deep Double Deterministic Policy Gradient (SD3) achieves superior performance over Twin Delayed Deep Double Deterministic Policy Gradient (TD3) [11] on continuous control reinforcement learning tasks. We utilize both environments that were used by the paper and expand to include some not present.

**Methodology**

We compare the performance of TD3 and SD3 on a variety of continuous control tasks. We use the authors' PyTorch code but also provide Tensorflow implementations of SD3 and TD3 (which we did not use for optimization reasons). For the control tasks we utilize OpenAI Gym environments with PyBullet implementations, as opposed to MuJoCo, in an effort to bolster claims of generalization and to avoid exclusionary research practices. Experiments are conducted both on similar environments in the original paper and those that were not mentioned.

**Results**

Overall we reach similar, albeit much milder, conclusions as the paper, specifically, that SD3 outperforms TD3 on some of continuous control tasks. However, the advantage is not always as readily apparent as in the original work. Algorithmic performance was comparable on most environments, with SD3 providing limited evidence of definitive superiority. Further investigation and improvements are warranted. The results are not directly comparable to the original paper due to differences in physics simulators. Additionally, we did not perform hyperparameter optimization, which could potentially bolster returns on some environments.

**What was easy**

The authors' made their code extremely easy to use, run, modify and rewrite in a different package. Because everything was available on their github and required only common reinforcement learning packages it was quick and painless to run. It was trivial to use the algorithms on different environments from different packages and collect their results for analysis.

**What was difficult**

One of the biggest difficulties was the time and resource consumption of the experiments. Running each algorithm on each environment with a sufficient number of random seeds took the vast majority of the time. We had a total runtime of around 310 GPU hours (or 13 days). Time was our primary constraint and was the primary reason we did no investigate other environments. Simulator differences also proved to be somewhat challenging.

**Communication with original authors**

Our contact with the authors was limited to a discussion we had at their poster presentation at NeurIPS 2020.

---

# 1 Introduction

Deep reinforcement learning (RL) has achieved a great deal in the past decade. From mastering games such as Go [27], Dota 2 [5], StarCraft II [32], and Atari [4], to precision robotic control [2] and robotic movements [14]. However, there is still substantial room to grow, and RL suffers from a number of problems. Problems such as brittleness to hyperparameters [17] and small code level changes [9], inferior performance to far simpler methods [12], [23], and weak generalization [21] that ultimately make RL difficult to use in any real world applications [8]. This problem of generalization is key to the investigation presented here. In this work, we attempt to evaluate the generalizability of the novel continuous control reinforcement algorithm presented in Pan et al. [25], the Softmax Deep Double Deterministic Policy Gradient (SD3).

Pan et al. [25] presents an empirical and theoretical argument for the usage of the softmax operator in continuous control reinforcement learning tasks. While the softmax operator is standard practice in discrete policy gradient algorithms, it usage in continuous environments is rare. However, with the recent successes of entropy maximizing RL [10], [13], [14], [15], [16], [7], [33], there has been an increase of interest in the softmax operator for RL [3], [28] which is functionally similar to entropy maximization. SD3 continues this trend, utilizing the softmax operator to expand upon and claim improvements over TD3 [11] on the MuJoCo benchmark.

In this work, we attempt to evaluate the generalization of Pan et al. [25] utilizing a variety of environments based in the open source PyBullet physics simulator. These environments include PyBullet reimplementations of the environments from the original paper, in addition to some that were not present. These environments are chosen to test SD3's ability to generalize to other environments. These test environments utilize the PyBullet physics simulator and are adapted for OpenAI Gym [6] via PyBullet-Gym and includes some similar environments from the MuJoCo benchmark and many unique ones. We use PyBullet over MuJoco to support efforts to make reinforcement learning more equitable [24], and we reject exclusionary MuJoCo usage, conducting all of our experiments exclusively on free and open source software. Note that all code and results will be available for a final copy (but are not presented here to preserve anonymity).

# 2 Preliminaries

## 2.1 Reinforcement Learning Background

Reinforcement learning is a field of machine learning in which an agent seeks to maximize a numerical reward signal from an environment [29]. RL environments are often formalized as a Markov Decision Process (MDP), defined by the tuple $\langle \mathcal{S}, \mathcal{A}, R, \gamma \rangle$. Here $\mathcal{S}$ represents the set of states, $\mathcal{A}$ the set of actions, $R$ the reward and $\gamma$ the reward discount, $\gamma \in [0, 1]$. It is common to also see a $P$ in this tuple representing the probability of state transitions; however, our environments are not stochastic and therefore $P = 1$. The goal of an RL algorithm is to design (or learn) a policy, $\pi$, such that it maximizes the expected return (also called objective): $J = \mathbb{E}\left[\sum_{t=0}^{T} \gamma^t r(s_t, a_t)|\pi\right]$.

There are a number of techniques to design the policy $\pi$, but in contemporary RL it is standard practice to use non-linear function approximators (i.e. deep neural networks) to learn value and policy functions. While there are a number of classes of algorithms, the focus of this work is on model-free, off-policy, actor-critic algorithms. In these systems an actor (or policy) network outputs the actions and is updated with a critic network that learns the value function. These policy functions can either be stochastic or deterministic, i.e. they can either output the mean and standard deviation of the policy distribution for an action to be chosen from, or output a single value for the action. Both SD3 and TD3 are deterministic policy algorithms. The policy is network, parameterized by $\theta$, is denoted as $\pi_\theta$. This allows us to represent the objective function $J$, as an expectation, $J(\pi_\theta) = \int_{\mathcal{S}} r(s, \pi_\theta(s))ds = \mathbb{E}\left[R(s, \pi_\theta(s))\right]$, taking the gradient of the objective function yields $\nabla J(\pi_\theta) = \int_{\mathcal{S}} \nabla \pi_\theta(s)\nabla Q(s, \pi_\theta(s))ds = \mathbb{E}\left[\nabla \pi_\theta(s)\nabla Q(s, \pi_\theta(s))\right]$, where $Q(s, a)$ indicates the expected return of taking action $a$ in state $s$ [26]. This is known as the deterministic policy gradient (DPG). There are a variety of techniques and improvements to this formulation building upon each other to achieve better results. For a visual outline of this see Figure 1. The x-axis is time, and reading this figure left to right shows how the algorithms have built upon each other over time. Each box is an algorithm and the arrow from one box indicates that the pointed to algorithm took ideas and techniques from the previous algorithm.

DPG algorithm is essentially the same as described above, the policy is updated via the gradient above and the Q network is updated with the standard Bellman error: $\mathcal{L}(Q_\theta) = r_t + \gamma Q_\theta(s_{t+1}, \pi_\theta(s_{t+1})) - Q_\theta(s_t, a_t)$ [26]. However, DPG is very hard to train suffering from severe hyperparameter brittleness and lack of exploration. DDPG improved upon DPG by adding noise to the policies to increase exploration, and by adding target networks for the policy and value networks to improve stability [11]. DDPG still suffers from over-estimations of the Q value, which happens because overestimation is selected for when updating the Q value, i.e. $\mathbb{E}\left[max_a Q(s_t, a)\right] \geq max_a \mathbb{E}\left[Q(s_t, a)\right]$, often leading to convergence problems [20]. TD3 attempts to address the well know overestimation problem in Q learning.

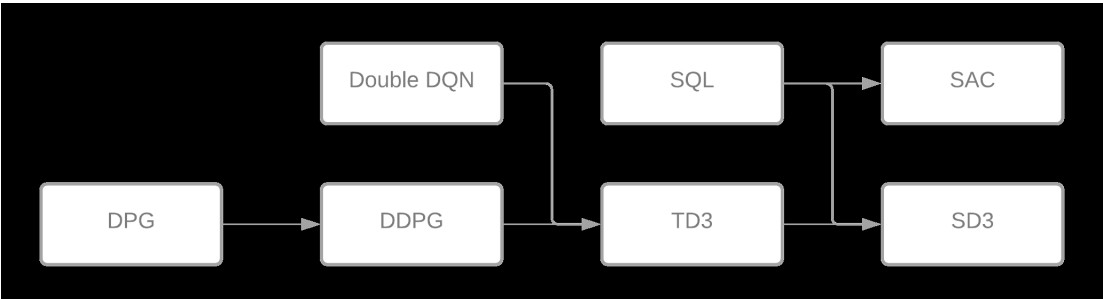

Figure 1: Outline of Continuous Control Algorithms. Deterministic Policy Gradient (DPG) [26], Deep Deterministic Policy Gradient (DDPG) [22], Double Deep Q Networks (Double DQN) [31], Twin Delayed Deep Deterministic Policy Gradient (TD3) [11], Soft Q Learning (SQL) [13], Softmax Deep Double Deterministic Policy Gradients (SD3) [25], Soft Actor-Critic (SAC) [16]

To address these overestimation errors, TD3 borrows from Double DQN [31] and uses the idea of using two (hence the name 'twin') Q function approximators to prevent overestimations, in addition TD3 updates the policy network less frequently [11]. While TD3 does successfully address the overestimation problem, it introduces the new underestimation problem, something SD3 tries to combat [25]. Parallel to these developments are the improvements in entropy maximization methods: soft Q Learning [13] and soft actor-critic [15]. These methods maximize a different objective than presented above due to the addition of an entropy term: $J(\pi) = \sum_{t=0}^{T} \mathbb{E}\left[r(s_t, a_t) + \alpha \mathcal{H}(\pi(s_t))\right]$. This entropy objective (under optimal conditions) is functionally the same as the softmax operator.

## 2.2 SD3

SD3 has a number of similarities with TD3, the main difference being the use of the softmax operator in the value function Bellman error. Hence, key to understanding SD3 is understanding the softmax operator. The softmax operator is common in RL problems, but is typically seen in discrete action spaces where it is easy to calculate: $\sigma(z)_i = \frac{e^{z_i}}{\sum_{j=0}^{K} e^{z_j}}$. If we consider the softmax of the Q function, in continuous action spaces it becomes computationally intractable: $\sigma(Q(s, \cdot)) = \int_{\mathcal{A}} \frac{exp(\beta Q(s,a))}{\int_{\mathcal{A}} exp(\beta Q(s,a'))da'} Q(s,a)da$. If we could utilize the softmax operator, Pan et al. [25] proves helpful bounds on the difference between $max_a Q(s, a)$ and $\sigma(Q(s,a))$, showing that the softmax operator does not overestimate and worst case only slightly underestimate. In order to make the continuous softmax operator computationally feasible, SD3 utilizes a sampling technique from [16]: $\sigma(Q(s, \cdot)) = \mathbb{E}\left[exp(\beta Q(s,a))Q(s,a))\right] / \mathbb{E}\left[exp(\beta Q(s,a))\right]$. This sampling technique can also incorporate advancements in importance sampling, but that is unused in SD3. To illuminate the similarities and differences between the TD3 and SD3 we present them side by side highlighting a few key differences (blue are highlighted in both to show differences and red are only highlighted in one to show addition of a new feature). See Algorithms 1 and 2.

# 3 Methodology

## 3.1 Target Questions

In order to provide a thorough assessment of the paper, the claims it makes, and the conclusions it draws, we present three central questions. These questions serve to guide our analysis, and we evaluate them after presenting the results.

- To what extent can we replicate the superior performance of SD3 over TD3 on PyBullet reimplementations of the used MuJoCo environments?
- To what extent does this performance generalize to other continuous control tasks?
- What improvements can be made to the SD3 algorithm?

## 3.2 Experimental Setup

Although we provide TensorFlow [1] implementations of both TD3 and SD3, we run all experiments using the authors' provided PyTorch implementations. Our reasoning is twofold. First, our code is less optimized than the PyTorch code and it is therefore more computationally feasible to use the PyTorch code. Secondly, RL algorithms are notoriously

difficult to re-implement [30] and we wish to avoid any challenges to our implementations. Even little differences, such as rounding vs. truncating floating points, can result is performance differences. We wanted to make our claims about the algorithm as presented, and our claims are strengthened by utilization of their code. The PyBullet gym adaptions and implementations can be found here. We begin by collecting data for the 6 of the environments in the original paper, specifically: Ant, Hopper, Lunar Lander, Walker2D, Humanoid, and Half Cheetah environments. We also evaluated three additional environments: Pendulum, InvertedDoublePendulum and HumanoidFlagrun (one of the hardest environments). Our experiments runs were setup using the same framework as the original paper: we collected data for 1 million iterations and repeated each experiment five times with a different random seed each time. For the extended experiments we only collected three runs due to time constraints. All experiments were conducted on two personal computers with CUDA enabled GPUs. Depending on the environment each run would take between 2 - 8 hours.

---

**Algorithm 1:** TD3

Initialize value networks $Q_1, Q_2$ with parameters $\theta_1, \theta_2$
Initialize policy network $\pi$ with parameters $\phi$
Initialize target networks $Q_1', Q_2', \pi'$ with parameters $\theta_1' \leftarrow \theta_1, \theta_2' \leftarrow \theta_2, \phi' \leftarrow \phi$
Initialize replay buffer $\mathcal{D}$
**for** *for t = 0 to T* **do**
  Select noisy action $a \leftarrow \pi(s) + \mathcal{N}$ and observe reward and new state $s'$
  Store $\langle s, a, r, s' \rangle$ in $\mathcal{D}$
  Randomly sample N tuples from $\mathcal{D}$
  $y \leftarrow r + \gamma min_{i=1,2} Q_{\theta_i'}(s', \pi_\phi(s') + \mathcal{N})$
  Update critics via the loss
  $\mathcal{L} \leftarrow \frac{1}{N} \sum (y - Q_{\theta_i}(s,a))^2$
  **if** *policy update* **then**
    Update $\phi$ via gradient
    $\frac{1}{N} \sum \nabla \pi_\phi(s) \nabla Q_{\theta_1}(s, \pi_\phi(s) + \mathcal{N})$
  $\theta_i' \leftarrow \tau \theta_i + (1-\tau)\theta_i'$
  $\phi' \leftarrow \tau \phi + (1-\tau)\phi'$

---

**Algorithm 2:** SD3

Initialize value networks $Q_1, Q_2$ with parameters $\theta_1, \theta_2$
Initialize policy networks $\pi_1, \pi_2$ with parameters $\phi_1, \phi_2$
Initialize target networks $Q_1', Q_2', \pi_1', \pi_2'$ with parameters
  $\theta_1' \leftarrow \theta_1, \theta_2' \leftarrow \theta_2, \phi_1' \leftarrow \phi_1, \phi_2' \leftarrow \phi_2$
Initialize replay buffer $\mathcal{D}$
**for** *for t = 0 to T* **do**
  Select action $a \leftarrow \pi_i(s), i \leftarrow max_{i=1,2} Q_i(s, \pi_1(s))$
    and observe reward and new state $s'$
  Store $\langle s, a, r, s' \rangle$ in $\mathcal{D}$
  **for** *i = 1, 2* **do**
    Randomly sample N tuples from $\mathcal{D}$
    Sample K noises $\epsilon$
    $\hat{a}' \leftarrow \pi_{\theta_1'}(s) + \epsilon$
    $\hat{Q} \leftarrow min_{i=1,2}(Q_{\theta_i'}(s', \hat{a}'))$
    $\sigma(\hat{Q}) \leftarrow exp(\beta\hat{Q}(s', \hat{a}'))\hat{Q}(s', \hat{a}')/exp(\beta\hat{Q}(s', \hat{a}'))$
    $y \leftarrow r + \gamma\sigma(\hat{Q})$
    Update $Q_{\theta_i}$ via the loss $\mathcal{L} = \frac{1}{N} \sum (y - Q_{\theta_i}(s,a))^2$
    Update $\phi_i$ via gradient
    $\frac{1}{N} \sum \nabla \pi_{\phi_i}(s) \nabla Q_{\theta_i}(s, \pi_{\phi_i}(s))$
  $\theta_i' \leftarrow \tau \theta_i + (1-\tau)\theta_i'$
  $\phi_i' \leftarrow \tau \phi_i + (1-\tau)\phi_i'$

## 3.3 Hyperparameters

We use the same hyperparameters as the original paper. For all environments and algorithms refer to Table 1.

| | |
|---|---|
| Batch size | 100 |
| Network architecture (policy and value) | (400,300) |
| ADAM [19] learning rate | $1 * 10^{-3}$ |
| Replay buffer size | $1 * 10^6$ |
| Training delay | $1 * 10^4$ |
| Noise, $\mathcal{N}(\mu, \sigma^2)$ | $\mathcal{N}(0, 0.1)$ |
| $\gamma$ | 0.99 |
| $\tau$ | 0.005 |
| Policy update frequency (TD3 only) | 2 |
| K (SD3 only) | 50 |

Table 1: Hyperparameters

The one notable difference is that Pan et al. [25] uses a separate set of hyperparameters for the Humanoid environment which we do not do. The second important note on hyperparameters is the SD3 unique hyperparameter $\beta$, which scales the softmax operation. In the original paper this is determined to be a specific value for each environment, ranging from 0.001 to 500. On the environments utilized in the paper we use the same values of $\beta$. For the extended environments we adopt the values of $\beta$ from similar environments. For all $\beta$ value see Table 2

| Ant | 0.001 |
|---|---|
| Half Cheetah | 0.005 |
| Hopper | 0.05 |
| Lunar Lander | 0.5 |
| Walker 2D | 0.1 |
| Humanoid | 0.05 |
| Pendulum | 0.5 |
| Inverted Double Pendulum | 0.5 |
| Humanoid Flagrun | 0.05 |

Table 2: $\beta$ Values

## 4   Results

Our results are overall indicative that SD3 does provide an advantage on the some of the environments, although these advantages are relatively small. On all 9 environments, SD3 performers an average of 7.7% better than TD3; however this comes at an increased computational cost and is not consistently superior.

### 4.1   Results on Paper Benchmarks

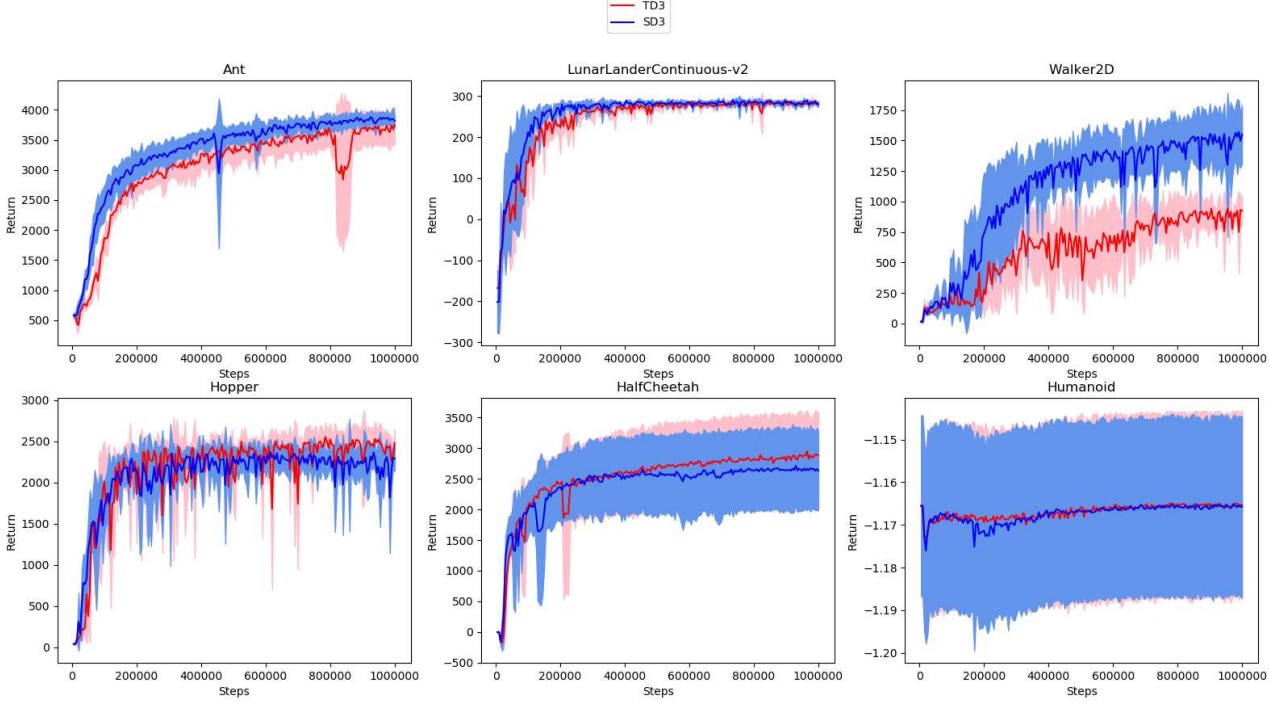

Figure 2: Paper Environments Reward vs Million Steps

Results for six of the original environments can be seen in Figure 2. The blue represents SD3 and red TD3. The shaded area represents a confidence interval of one standard deviation. While the exact numerical rewards of the PyBullet are not directly comparable[1] to the MuJoCo rewards (hence we cannot overlay the original papers results directly on these graphs); the environments are evaluate the same goal and the same physics. The results are also presented in Table 3. This table shows the best average reward (over 5 runs) and the associated standard deviation. The better performing result is bolded. From these results we can see that SD3 outperforms TD3 on 2/6 of the environments. This may look as though these algorithms are effectively the same (as the humanoid performance difference in minuscule).

---

[1]The reward scale is lower than MuJoCo

| Environment | TD3 | SD3 |
|---|---|---|
| Ant | $3744.6 \pm 305.5$ | $\mathbf{3878.3 \pm 103}$ |
| HalfCheetah | $\mathbf{2948.1 \pm 665.7}$ | $2711.3 \pm 644.8$ |
| Hopper | $\mathbf{2553.1 \pm 181.2}$ | $2367.3 \pm 157.8$ |
| LunarLander | $\mathbf{290.1 \pm 4.6}$ | $289.7 \pm 5.5$ |
| Walker2D | $942.4 \pm 132.8$ | $\mathbf{1572 \pm 260}$ |
| Humanoid | $\mathbf{-1.1649 \pm 0.022}$ | $-1.1651 \pm 0.021$ |

Table 3: Paper Environments Comparisons

However, further analysis gives a slight edge to SD3. The average performance of SD3 is 9% better than TD3[2] on these environments. This is very minor improvements (and not a reliable one) and comes at the cost of increased computation time. Walker2D is the only environment that one could universally recommend SD3 over TD3 as in every other environment the standard deviation curves overlap. There seems to be consistent failues in the Humanoid PyBullet environment, which both algorithms fail to learn.

## 4.2 Additional results not present in the original paper

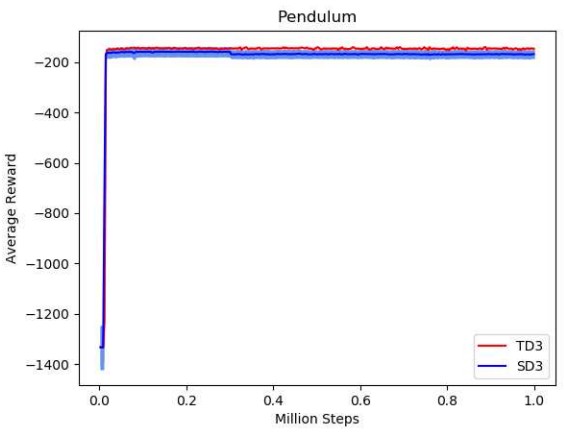

(a) Pendulum Reward vs Million Steps

(b) Double Inverted Pendulum Reward vs Million Steps

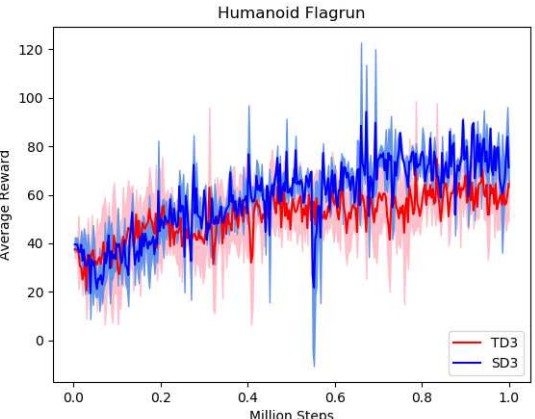

Figure 3: Humanoid Flagrun Reward vs Million Steps

| Environment | TD3 | SD3 |
|---|---|---|
| Pendulum | $\mathbf{-139.2 \pm 0.9}$ | $-157.4 \pm 18.1$ |
| Inverted Double Pendulum | $\mathbf{9358.4 \pm 0.6}$ | $9357.1 \pm 0.1$ |
| Humanoid Flagrun | $74.8 \pm 23$ | $\mathbf{94.2 \pm 19.4}$ |

Table 4: New Environments Comparisons

Standard benchmarks are often problematic and may not always generalize (a reinforcement learning of "teaching for the test" so to speak). While there are proposals for other metrics of evaluation [18], we choose to do a simple test for generalization by expanding the testing on environments that are not part of the "standard" benchmark. These These three environments offer similar results to the paper environments, a positive indication for generalization. SD3 performs worse on the majority of the environments once again, while also performing (on average) 5% better[3]. Note that this is without substantial hyperparameter ($\beta$) optimization. This is indicative that even without full knowledge of hyperparameters, reasonable choices can lead to an advantage. The point of these extended results is less about a direct comparison These results are less about the potential strength of SD3 compared to TD3, and more about the generalizability and out of the box usability.

# 5 Discussion

## 5.1 Results Analysis

The results presented above are not entirely conclusive. Although they indicate that, on average, SD3 performs superior to TD3, this is not the end all be all. SD3 only outperforms TD3 on 3 out of the 9 environments, but it outperforms TD3 substantially (hence why the average is in its favor). Note too that SD3 is a more computationally expensive algorithm (only by a small margin, comparable to its performance gains). However, the runtime of the algorithm does not include the necessary hyperparameter optimization for the $\beta$ value that would be needed (which could require 5-10 additional runs) for any real world applications. The advantages appear to be more nuanced than the original paper suggested.

## 5.2 Target Questions

Next, let us consider the target questions we set in the beginning of this work. To what extent can we replicate the superior performance of SD3 over TD3 on the given environments? In short, we we largely not able to. We were able to replicate superior performance on some environments and on average, but the majority of environments we were not able to. In addition, the size of the advantages present in the original paper do not appear to be as large here. Pan et al. [25] presents 4 environments that SD3 definitively (i.e. the standard deviation curves do not overlap) performs better on: Half Cheetah, Ant, Walker2d, Hopper. However, we were only able to replicate this level of superior performance on Walker2d. On the other three environments, SD3's advantage was minimal to nonexistent. This is not to suggest that there is anything wrong with their results, rather, that the results may not generalize (even to extremely similar environments, with minorly different reward scaling). This is unfortunate, as decreasing the brittleness of RL algorithms is necessary for real world applications.

To what extent does this performance generalize to other continuous control tasks? As was mentioned above, this generalization is weak. The generalization is minimal even to the same environments in a different physics simulator and this is also true for the new environments. We cannot say definitively that SD3 is the inferior algorithm on Pendulum and Double Inverted Pendulum as we did not do the full ablation studies to determine the optimal $\beta$ values. However, we can say that if one wants SD3 to perform definitely better, specific values of $\beta$ are needed, and even then performance may not be superior to TD3. This need for hyperparameter optimization is a weakness of the algorithm. Given the already numerous challenges of real world RL [8], requiring extensive trial and error to obtain the necessary parameters for algorithmic superiority is a steep price. Given the results on a new simulator and new environments, the SD3 does not appear to generalize particularly well. Of course, this is the case for many algorithms (and is not necessarily a unique flaw of SD3).

What improvement can be made to the SD3 algorithm? Although improving generalizability is an important problem, solutions are much more difficult. However, one solution that would help would be to enable automatic adjustments of the $\beta$ value. This is not only prohibitive when using SD3 for new environments (as hyperparameter optimization

---

[2](3878.3/3744.6 + 2711.3/2948.1 + 2367.3/2553.1 + 289.7/290.1 + 1572/942.4 + 1.1649/1.1651)/6 = 1.09

[3](139.2/157.4 + 9357/9358 + 94.2/74.8)/3 = 1.05

can be expensive) but also over the course of a single run, the optimal $\beta$ might differ. This idea is very similar to the improvements made to Soft-Actor Critic. In the original paper, the entropy parameter, $\alpha$, was determined via trial and error [15]; however, automating this parameter showed to be more effective from both a computation expense and a maximum reward standpoint [16]. The exact technique may not be able to carry over, as the nature of the problems are different, but it is certainly worth investigating.

## 6    Conclusion

In this work, we evaluate the replicatability of the paper Softmax Deep Double Deterministic Policy Gradients [25]. To promote inclusive research practices, we ran all code on the open source PyBullet physics engine. Our results generally align with the original paper's claims about SD3's superior performance over TD3 overall, are not very compelling. SD3 failed to offer an advantage on the majority of environments evaluated. However, the average performance boost warrants further investigation and there is potential that hyperparameter optimization would bolster the performance of SD3. It is worth noting that this level of scrutiny is not applied to all algorithms, and we make no claims that other SotA continuous control algorithms would generalize any better.

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
