# OpenReview forum: "Replicating Softmax Deep Double Deterministic Policy Gradients"
_ML_Reproducibility_Challenge/2020 — Reject_

### Official Review · AnonReviewer2 · 2021-02-27
**Clearly written report, with underwhelming findings**

**Rating:** 4
**Confidence:** 4

**Review:**

This work is an effort towards reproducing « Softmax Deep Double Deterministic Policy Gradients » by Pan et al. (2020). This algorithm (SD3) claims superior performance compared to TD3 by using the softmax operator during bootstrapping, instead of the conservative « min » rule used in TD3. The main finding of this reproducibility report is that there seems to be indeed a small benefit of SD3 vs TD3, but it is not as marked as in the original paper, and it was even less clear on new environments.

Overall the report is clearly written and easy to follow. My main criticism is that, as far as I can understand, the authors mostly re-used the existing open source implementation from Pan et al., and applied it to PyBullet environments (vs. the original MuJoCo ones). There is no hyper-parameter search nor ablation study. I do sympathize with the lack of computational resources, but maybe a different work should have been selected if the authors did not have enough to dig deeper. This makes the overall contribution somewhat limited, especially since the authors did not try to use the same MuJoCo environments, so it is not 100% clear if the differences (vs. the original paper) only comes from the environments themselves, or also from other (unknown) factors. The authors do mention they will provide a TensorFlow re-implementation of the algorithm, but since it wasn’t used for the report, is said to be less efficient than the publicly available PyTorch implementation, and there is no mention on whether or not it can reproduce the same results, I am not sure that it brings added value here.

In the end, it is not clear if the discrepancy vs. the original paper is because SD3 is not that good compared to TD3 on the PyBullet environments, or because hyper-parameters need to be adjusted for these environments, or because a mistake was made somewhere. This makes it difficult to draw meaningful conclusions from this report.

Minor detail: on l. 88, should max_q be max_a?

**Familiar With The Original Paper:**

I have not read the original paper

**Reproducibility Summary:**

Report has summary

---

### Official Review · AnonReviewer3 · 2021-03-02
**Review of "Reproducing Softmax Deep Double Deterministic Policy Gradients"**

**Rating:** 5
**Confidence:** 4

**Review:**

Overall, this report represents a reasonably thorough reproduction of the Pan et al. paper. However, there are some weaknesses (as outlined below), that should be improved.

Strengths
- The suggestion on how to improve the performance of SD3 (lines 160-163) is interesting, and could lead to some non-trivial improvements.
- Evaluation the results in a set of comparable open-source environments is a good goal, and aids reproducibility overall.
- Highlighting the differences between TD3 and SD3 is excellent for exposition and clarity.
- This report is written in a way that makes it reasonably clear even without familiarity with the original paper.

Weaknesses
- One reason that the authors suggest for a discrepancy between the rewards in the Pan et al. paper and their reproduction is their use of PyBullet instead of MuJoCo (lines 127-128, lines 153-154). However, this claim is not validated (or cited, other than the referenced GitHub issue).
- What do the edges in Figure 1 signify? Presumably that the algorithm uses ideas from an earlier one, but this should be clarified in the text.
- While the authors state that the results achieved are weaker than those in the original paper, there is no quantitative evidence presented, other than some graphs. What is the magnitude of the effect? What is its statistical significance?
- While the RL background section (Section 2) is appreciated, there are some inaccuracies, including "RL is often formalized as a Markov Decision Process": it is the agent's interactions with the environment that are formalized by the MDP, but RL itself is the process of learning a good policy in that environment.
- The report requires substantially more proofreading. Examples include: "author's" (line 21), "consumption's" (line 26), "reprehensibility" instead of "reproducibility" (line 165), lack of consistent capitalization throughout.


**Familiar With The Original Paper:**

I have read the original paper

**Reproducibility Summary:**

Report has summary

---

### Decision · Program_Chairs · 2021-03-31

**Decision:**

Reject

**Comment:**

Overall reviews and/or the paper content not good enough for the AC to recommend to the journal.